# Mindfulness as a Protective Factor for Dissatisfaction in HCWs: The Moderating Role of Mindful Attention between Climate Stress and Job Satisfaction

**DOI:** 10.3390/ijerph17113818

**Published:** 2020-05-28

**Authors:** Tiziana Ramaci, Venerando Rapisarda, Diego Bellini, Nicola Mucci, Andrea De Giorgio, Massimiliano Barattucci

**Affiliations:** 1Faculty of Human and Social Sciences, Kore University of Enna, 94100 Enna, Italy; 2Occupational Medicine, Department of Clinical and Experimental Medicine, University of Catania, 95121 Catania, Italy; vrapisarda@unict.it; 3Department of Business Economy, University of Verona, 37129 Verona, Italy; diego.bellini@univr.it; 4Department of Experimental and Clinical Medicine, University of Florence, 50134 Florence, Italy; nicola.mucci@unifi.it; 5Faculty of Psychology, e-Campus University, 22060 Novedrate (CO), Italy; andrea.degiorgio@uniecampus.it (A.D.G.); massimiliano.barattucci@uniecampus.it (M.B.)

**Keywords:** climate stress, dispositional mindfulness, job control, satisfaction, HCWs, mindful attention

## Abstract

With the aim of investigating the possible moderating effect of job control and dispositional mindfulness between different sources of organizational stress and job satisfaction, a correlational study was designed involving health care workers (HCWs). The following questionnaires were administered and completed by 237 HCWs: (1) Occupational Stress Indicator (OSI), to measure the sources of stress at work (managerial role, climate power, climate structure, internal relationships), and job satisfaction; (2) Mindfulness Attention Awareness Scale (MAAS) to assess the individual’s level of attention to what is taking place in the present; (3) Job Control Scale (JCS) to assess the perceived control at work. Hierarchical regression analyses were used to test the hypothesized relationships between variables; the results showed that, between the different sources of stress, the organizational climate dimension was negatively associated with job satisfaction; moreover, mindfulness attention moderated the relationship between climate stress and job satisfaction; unexpectedly, the interaction between job control and the organizational climate dimension was not significant in affecting job satisfaction. This study can provide useful information for Human Resources Management (HRM) practices regarding job and mental control interventions and empowerment, and possibly offer a new interpretation of the role of attention to what is happening in the present moment and autonomy between climate stressors and occupational satisfaction.

## 1. Introduction

Stress is a global issue in modern life; according to the current definitions, stress is a process involving appraisal, response, and attempts to cope with and manage stressors in order to meet goals [1,2]. 

Profound reforms and changes in health care systems, combined with typical organizational dynamics (e.g., environmental conditions, required quality standards and pressure, role conflicts, limited autonomy, etc.), make hospitals among the workplaces more exhausting and demanding in any cultural and dimensional context [3]. Organizational, hierarchical, and relational work-related stressors in the health sector impact negatively on health care workers’ (HCWs’) mental and physical condition, causing consequences and costs to the patients, the organization, and society [4]. 

Institutional references from different backgrounds and nationalities are common in indicating the importance of developing and implementing practices and interventions to support the HCWs and the organization in stress management [5]. 

Literature results regarding organizational stress factors have led to the differentiation of two types of stressors at work: (1) those concerning circumstantial tasks (e.g., violent clashes, accidents, etc.), and therefore related to critical service events; (2) those concerning the work context, e.g., organizational context, cultural norms, environment, relationship with superiors, and therefore related to organizational aspects [6]. 

The literature that focused on identifying the organizational factors that are most linked with negative organizational perceptions and outcomes considered, overall, five major sources of occupational stress: roles of the individual in the organization, career development, in-working relationships and, organizational structure and climate [7,8]. 

Furthermore, between the different organizational factors, perceptions of job control seem to clearly help employees to experience positive responses, work outcomes, attitudes, and satisfaction [9,10,11].

Other contributions focused more on psychological factors and differences in reactions to stress [12], taking into account the importance of the subjective reactivity in physiological and psychological response, and of some specific psychological skills (e.g., coping, resilience tolerance, etc.) [13]. 

In this vein, several studies explored the role of different stress management tools [14,15], and among all, those mindfulness-based received a great interest [16], targeting the interest for the role of awareness and attention in stress perceptions and consequences. 

Despite the numerous studies that have explored the effects of job control and mindfulness-based programs on work outcomes [17,18], to the best of our knowledge, no studies investigate the role of dispositional mindfulness (DM) on these outcomes. Literature described the DM as the keen awareness and attention to our thoughts and feelings in the present moment [19]. In regard to emotional regulation, DM is considered by literature both an intraindividual protective factor for stress events and as an adaptive strategy for regulating negative emotions through observing the ongoing emotion/thoughts with a non-judgmental attitude [19]. This is not strictly due to the mindfulness practice but is considered an intrinsic psychological resource that allow to the individuals to naturally pay attention to the present moment, allowing these people to recover quickly from negative emotions [20,21]. Working environments can be particularly stressful and negative emotions can emerge by them; people with a high DM level are able to reducing negative emotions, also related to stressful events [22,23].

With the aim of investigating the possible moderating effect of job control and DM between different sources of stress and job satisfaction, a correlational research was designed involving health care workers (HCWs).

The results of the study can provide useful information for Human Resources Management (HRM) practices regarding the job and mental control interventions and empowerment, and possibly offer a new interpretation of the role of mindfulness and autonomy between stressors and job satisfaction [24].

## 2. Job Control, Organizational Stress, and Work Outcomes

Adaptation is significantly dependent on the worker’s assessment of the stressful event since stress occurs when there is maladaptation between the person and the environment in terms of demands and resources [25].

Extending recent literature underlines the notion that control over work within the workplace is regarded as one of the most important elements of the stress process, and it is the focus of many theories on work stress [26,27]. Job control is defined as a perceived ability to exert some influence over one’s work environment, in order to make it more rewarding and less threatening, and constitutes an individual’s belief in his/her ability to affect the desired change on their work environment. Some contributions envisioned control as one’s degree of autonomy or decision authority over tasks, including ordering of task completion, discretion in how tasks are completed, or a degree of autonomy over the nature of the tasks themselves [28,29].

Theories of occupational health and performance have hypothesized that providing people with control over their work helps to improve job satisfaction and well-being.

In line with these theories [30,31] researchers showed that, apparently, there is consistent evidence that high levels of work control are associated with a higher level of satisfaction and lower levels of stress. Specifically, the ability to intervene and change work processes may reduce inherently stressful cognitions of having insufficient resources to complete tasks. This is corroborated by numerous studies that have found strong negative associations between perceived control and other measures of psychological strains. Perceiving a lack of control is proved to induce strain by frustrating the intrinsic need to feel satisfied [32,33,34], while demanding work can engender positive outcomes when accompanied by high levels of control [28]; and this appears particularly relevant in jobs (as HCWs) characterized as being “high strain” due to a lack of job control [30,31]. De Jonge and collaborators found a positive relationship between sources of stress and job satisfaction for those who reported high levels of control, and a negative relationship for those reporting low control [35]. Employees who perceive control over their environment assess the demands of their jobs more accurately and cope with them positively, experiencing optimistic responses, work outcomes, attitudes, and satisfaction [35,36].

The literature review seems to indicate, overall, that job control can act as a moderating factor in helping workers cope with stressors at work and be more satisfied. Investigating these aspects can have a great applicative value, through the implementation of organizational intervention practices on autonomy, decision making, and control of the worker’s jobs.

## 3. Dispositional Mindfulness and Job Outcomes

Mindfulness is defined as *“a state of consciousness characterized by receptive attention to and awareness of present events and experiences, without evaluation, judgment, and cognitive filters”* [37]. Generally, mindfulness can be seen as a personal resource that enables people to cope with the demands by helping them focus their attention on the present moment rather than concentrating on problems and consequences beyond their control [38,39]. Mindfulness can be improved through practice (e.g., MBSR, mindfulness-based stress reduction [40]) or, as aforementioned, can be an intrinsic psychological resource [41]. Employees who pay too much attention to internal events, e.g., a low DM level with a judgmental attitude, engage in less avoidant behavior amongst dissatisfaction [42].

Referring to a large strand of studies that focused on interventions at work for stress management, mindfulness-based practices seem to play a crucial role in the stress process [43,44,45,46,47,48]; many studies suggest that the mindful control state helps people to separate environment characteristics from the corresponding reactions, limiting automatic responses to the environment, and focusing attention on one’s physiological responses [32,49,50]. It facilitates stress resilience and more positive coping [33,34,51] because it draws people into the present moment to help them experience greater control over the events they experience [52,53].

Mindful people, who have practiced or that have the disposition to be so, experience mastery in coping with difficulties. Indeed, research does suggest that if people can learn to focus on the task at hand (e.g., by learning acceptance or attention), then they are better able to notice and respond effectively to even subtle changes in contingencies of reinforcement (e.g., situations in which they have, and can use, control) [54,55,56]. These mechanisms seem to allow mindful individuals to cope more effectively with difficult events such as stress [57,58].

Mindfulness effectiveness in promoting positive outcomes at work seems to be related to the role of the practice in strengthening attention, compassion, awareness, and acceptance of emotions; emerging evidence seems to indicate that emotional intelligence can moderate perceived stress and affect [59,60,61]

There are several studies conducted in the nursing field: the pilot study led by Mackenzie (2006) showed a reduction in stress symptoms and improved well-being in nurses undergoing a mindfulness program [62]; Keehley and Abercrombie found an important reduction in anxiety and self-critical and judgmental thinking in nurses [63]; the most recent study conducted by Magtibay in 2017 focused the attention on work-related stress as a cause of burnout in nurses, poor patient care and a significant increase in costs for American healthcare companies [64].

Although many studies have focused on the role of mindfulness in the impact that sources of stress have on perceived stress, very few have tried to investigate the exact role between sources of stress and job satisfaction [65,66,67,68].

There is now greater consideration than before that stressors may be important in predicting influence reactions in satisfaction outcomes. Moreover, interest in the influence of DM on psychological outcomes has been gathering pace over recent years.

On the contrary, to date, research on DM in workplaces has not been systematically conducted in Italy.

Studies have largely neglected to investigate the role of factors associated with mindfulness patterns, which have been related to adaptive responses to heavy stressors, like those found in hospitals. Even less studies on this topic, focusing on the independent role of mindfulness and control on outcomes has not been conducted, in a no-clinical sample.

## 4. Study Aim and Hypotheses

Work-related stress is a problem. It entails a cost for society, and, therefore, companies are trying to explore which are the most influencing organizational stressors on relevant job outcomes (among all, job satisfaction) and to implement stress management strategies, especially aimed at healthcare workers and, most of all, HCWs.

HCWs belong to one of the categories of workers that are most at risk of work-related stress [69,70].

Following the literature review [18,61,71,72], the present study wanted to answer to the following research gaps and questions:Which is the right role of job control between organizational sources of stress and job satisfaction?Which is the role of DM in reducing the effect of organizational stress factors on job satisfaction?

Between organizational sources of stress, managerial roles, relationships with colleagues, organizational climate structure and power are proved to produce significant effects on worker satisfaction, behaviors, attitudes, and performance [73,74]. Furthermore, job control perceptions and DM proved to be effective in reducing the effect of these sources of organizational stress on job satisfaction. Perceptions of control over work and DM may play an important role in job satisfaction and other outcomes [29].

Taking into account results from the abovementioned studies and from a previous study [61] that investigated mindfulness disposition and flexibility as mediators between sources of stress (managerial power, relationships at work, and intrinsic factors) and outcomes in HCW (psychological and physical consequences), and considering literature review regarding organizational climate [73], a correlational study was designed to investigate the relationship between different organizational sources of stress (managerial roles, relationships with colleagues, and organizational climate— structure and power—and job satisfaction, in a sample of HCWs; more specifically, the study investigates the possible moderating effect of job control and dispositional mindfulness between those organizational sources of stress and job satisfaction

The following hypotheses are suggested:

**Hypothesis** **(H1).**
*Sources of stress (managerial roles, relationships with colleagues, organizational climate stress—structure and power—are negatively related to job satisfaction.*


**Hypothesis** **(H2).**
*Job control will moderate the relationship between sources of stress and job satisfaction.*


**Hypothesis** **(H3).**
*Dispositional mindfulness will moderate the relationship between organizational sources of stress and job satisfaction.*


The conceptual model is depicted in Figure 1.

## 5. Materials and Methods

### 5.1. Sample and Procedure

A project on “work-related stress and corporate well-being” concerning the staff of an emergency Hospital in Sicily was started in June 2019 and is still ongoing involving specific units.

The data collection has been carried out between July and September 2019 and was aimed at HCWs working in the above hospital. Nurses and doctors from around half of the wards in total provided readiness to participate in the research in the indicated period.

Qualified researchers invited a total of 237 HCWs from a Sicilian hospital of the National Health System, who took part in the study on a voluntarily basis (using an accessibility sampling), to complete a questionnaire during working hours. All participants signed the informed consent before taking part in the study.

Data collection was carried out with a total of 237 workers, 170 nurses (percentage of the total sample of nurses = 26.1%) and 67 doctors (percentage of the total sample of doctors = 19.8%); because, for the low response rate of the second sub-sample (*n* = 31; 46%), data analysis did not consider doctors; the rationale behind the identification or choice to use the homogeneous group of nurses is to conduct a precise analysis of a homogeneous group of workers’ subjects to the same organizational stressors.

The final sample of analysis was made up of 148 nurses (87% response rate), 33.1% (*n* = 49) were males and 66.7% were (*n* = 98) female. Their average age was 47.79 years (standard deviation (SD) = 8.02) ranging from 25 to 64. The tenure average was 16.6 (SD = 8.58) years. The number of working hours ranged between 1 and 12, and the average was 6.88 (SD = 1.23). The duration of employment was 16.67 (SD = 8.58) and ranged between one and thirty-eight years (Table 1).

All procedures performed in this study are in accordance with the 1964 Helsinki Declaration and its subsequent amendments or comparable ethical standards. The study was approved by the Ethics Review Board of Kore University.

Respondents were asked not to mention their name anywhere in the questionnaire to ensure anonymity. All data were managed according to the EU General Data Protection Regulation (GDPR).

### 5.2. Measures

The study was conducted using the following questionnaires:

The Occupational Stress Indicator (OSI) [75,76], which takes a combined person-situation approach to the conceptualization and measurement of occupational stress. This test is designed to detect a broad spectrum of psychosocial stress, within an organization, classified under four areas: occupational pressure or sources of stress, individual characteristics, control, coping, and consequences of occupational stress. A total of 167 items are rated on a 6-point response scale, ranging from 1 (absolutely false) to 6 (absolutely true). We re-phrased some of the OSI items to reflect direct statements.

For the purposes of this study, the following measures were investigated:

Sources of stress—sub-scale: managerial roles (MR: 7 items), relationships with other people (RF: 4 items), Climate—both organizational structure (SFs: 4 items) and power (SFr: 4 items).

Alpha assessment of scale reliability was, respectively, for MR (0.89), RF (0.74), SFs (0.65), and SFr (0.81); Mindfulness Attention Awareness Scale (MAAS) [77]. The MAAS is a 15-item scale designed to assess a core characteristic of dispositional mindfulness, namely open or receptive awareness of and attention to what is taking place in the present. It is related to, and predictive of, well-being constructs. Respondents are asked to rate how frequently they experience what is described in each statement using a 6-point Likert scale from 1 (almost always) to 6 (hardly ever), where high scores reflect a more mindful presence. An example of an item is “I find it difficult to stay focused on what is happening in the present.”

The coefficient alpha for awareness was 0.80 and for attention was 0.69.

The Job Control Scale (JCS) [26]. The 22-item Job Control Scale assesses a range of areas over which people can have control at work: the variety of tasks performed, the order of task performance, pacing, the scheduling of rest breaks, procedures and policies in the workplace, and the arrangement of the physical environment. Each item (e.g., “How much control do you personally have over the quality of your work?”) is rated on a 5-point Likert scale ranging from 1 (very little) to 7 (very much). Higher scores indicate greater levels of control. The psychometric properties of this scale appear good and reveal a single factor of control.

The coefficient alpha was 0.87.

Job Satisfaction Analysis scale (in Occupational Stress Indicator), composed of satisfaction at work (SJ) and satisfaction with the structure (SS) sub-dimensions (5 items); an example item is: “What do you think and how do you feel at work?”. Each item is rated on a 6-point response scale, using a range from 1 (“strong satisfaction”) to 6 (“strong dissatisfaction”). A high score indicates high job satisfaction.

The coefficient alpha was 0.88.

Socio-demographic variables—participants were asked to provide information on socio-demographic characteristics, such as gender (1 = male and 2 = female), age, and job-related variables, such as the length of service (seniority) and hours of service (Table 1).

### 5.3. Data Analysis

The research design was correlational. Descriptive analyses (mean, standard deviation, and Bravais–Pearson’s correlation coefficients) were computed to pre-investigate the data. An independent t-test and correlation analysis were performed to establish to what degree gender, years and working hours were associated with job satisfaction. Multiple regression analysis was performed to test Hp1 and our independent moderator hypotheses (Hp2–Hp3), through SPSS 21.0 (IMB Corps., Armonk, NY, US). To examine the moderating effect, job control and dispositional mindfulness were modeled as independent moderators between sources of organizational stress and job satisfaction. We included all the controlled variables into the first step of regression analysis. We added independent variables in the second step. Finally, in the third step, we added the interaction terms. The interaction term was computed by calculating the multiplication of the standardized moderators. We also standardized the independent and dependent variables.

Separate analyses were run with two sub-dimensions of mindfulness, either mindful awareness or mindful attention.

## 6. Results

### 6.1. Descriptive Statistics

Means (M), standard deviations (SDs), and bivariate correlations for the measured variables are shown in Table 2.

T-test analysis revealed gender differences for job satisfaction (t_145_ = 2.28, *p* = < 0.05; Male = 3.69, Female = 3.14). Multiple regression analysis showed that the length of service and working hours significantly predict job satisfaction, while age and working hours did not predict job satisfaction (Table 3).

### 6.2. Hypothesis Tests

Multiple regression analysis regarding hypothesis 1 highlighted that, between sources of stress, only organizational climate stress proved to negatively predict job satisfaction (Table 3).

Because of these results, the other sources of organizational stress (managerial role, climate power, and internal relationship) were excluded from the hierarchical regression analysis.

Analysis indicated that the interaction effects between mindfulness (awareness sub-dimension) and organizational climate structure stress were not significant, as reported in Table 4.

In reverse, the interaction between mindfulness (attention sub-dimension), and organizational climate stress was positive and significant (*p* < 0.05), as reported in Table 5.

The interaction is also depicted in Figure 2. Among nurses who perceived low and medium mindfulness (attention sub-dimension), there was a significant positive association between stress and job satisfaction. The respective tests of the simple slopes relating job satisfaction to stress at the low end and the mean of mindfulness resulted in β = −0.66, 95% confident interval (CI) = −0.97 to −0.35, t = −4.29, *p* <.001; and β = −0.41, 95% CI = 0.18 to 0.64, t = −3.60, *p* < 0.01. Conversely, for those who rated mindfulness relatively high, there was a non-significant association between stress and job satisfaction, β = −0.16, 95% CI = −0.49 to 0.18, t = 0.92, *p* = 0.357.

Hypothesis 2 was confirmed only for mindful attention.

Hierarchical regression analyses revealed that the interaction between job control and organizational climate structure stress was not significant, as reported in Table 4 and Table 5.

## 7. Discussion

Organizational stress is made up of different work environmental factors that act on individual worker’s perceptions and outcomes and can entail lots of costs for companies and individuals.

Sources of stress originate from the structural design and the characteristics of organizational processes and lead to negative effects on people or strong dissatisfaction [78].

The present research investigated the relationship between different sources of stress that affect job satisfaction (managerial roles, internal relationships, climate—both organizational structure and power—and the possible independent moderating effect of job control [28] and DM [31,39,48].

Between socio-demographic factors, gender, the length of service, and working hours contribute significantly to predict job satisfaction levels, while age does not explain that relationship.

Overall, the results show that, in the present sample, perceived occupational work climate is the only source of stress that predict job satisfaction in nurses, confirming literature highlights [78,79,80].

However, the data did not confirm the hypothesized moderating role of job control between organizational sources of stress and job satisfaction.

Mindfulness attention, but not awareness, proved to be framed as a possible moderator between organizational climate stress and job satisfaction.

Some studies showed that perceptions of control over work and mindfulness, by helping people focus their attention on the present moment rather than concentrating on problems and consequences beyond their control within the workplace [28], play an important role in individual stress and job satisfaction.

Employees who perceive personal control over their environment more accurately assess the demands of their jobs and cope with them positively and experience some optimistic responses, work outcomes, attitudes, and satisfaction.

These premises are not fully reflected in our data, since only mindfulness, particularly the attention dimension, seems to have a moderating effect on the satisfaction variable.

According to the literature, providing people with control over their work seems to improve job satisfaction, meant as a positive and pleasurable psychological state that results from the positive appraisal of one’s job [42,78,79], and mitigates the effects of sources of stress and their influence reactions in satisfaction outcomes. Nevertheless, in this study, job control does not seem to work as a moderator factor between organizational stress and job outcomes. We believe that job control does not vary enough to moderate the negative effect of climate on job satisfaction. Furthermore, many factors (e.g., resilience tolerance or coping, etc.) [28,29] could be involved in the interaction with job control in reducing the negative effect of job stress or into suppressing the moderating effect of job control.

This study contributed to the literature highlighting that DM can function as a personal resource moderating job satisfaction; more specifically, it seems that mindful attention can be considered a sort of hygienic factor capable of preventing the reduction in job satisfaction in nurses.

The results highlight several insights for practitioners and HRM; between the different sources of stress, organizational climate stress must be seriously monitored in order to intervene on workers’ satisfaction; management should work on reducing climate stress with appropriate interventions and a training program (communication, team building, decision making, etc.).

Therefore, a global organizational diagnosis as a synthesis of aggregated data according to important organizational variables is a particularly delicate point in the climate analysis process. This interpretation must be framed in a business context to characterize it and offer a meaning that can lead to effective actions to strengthen such climate.

For the climate analysis to be effective, one can act on the known aspects, elsewhere indicated, as well as on individual mindfulness attitude, which certainly affect the personal evaluation of the outcomes. Furthermore, it is essential to implement concrete actions, which may consist of training activities, organizational changes, internal communication programs.

Moreover, specific training on attention and individual control should be implemented, through skills development training and empowerment.

Mindfulness has been applied to different nursing contexts, specifically for the prevention of psychosomatic work-related stress disorders [41,80], and successfully conducted in different settings. An intervention that improves the overall psychological well-being might be more useful in encouraging a more adaptive approach to treat stress and its consequences along the health pathway [81,82,83]

Mindful employees can be more able to focus on immediate job demands, filter out extraneous job demands, enhance their ability to focus on utilizing essential job resources [78]. Workers who operate in demanding environments cope with those demands by using a variety of resources available to them [30,79], and mindfulness, as it can potentially reduce stress among employees, may have a role in enhancing job satisfaction. Therefore, for future interventions, it may be useful to first verify the DM using MAAS and then apply mindfulness protocols to improve awareness of the present moment in people who are lacking.

Indeed, according to recent theories on risk management, precise plans are needed to identify, analyze, evaluate, and control possible risks of work stress to prevent them or avoid their harmful effects [82]. Risk management is a recursive process: the initial identification of stress and risk is followed by a periodic review of the assessment and interventions, up to the optimization of organizational aspects that reduce the risk itself and increase the well-being of workers. This study has embraced the idea of measuring the company’s level of well-being, as well as the identification of individual factors, which, when cultured with targeted interventions, can offer improvements capable of affecting the overall functioning of the individual and the organization.

Some limitations to this study can limit the results’ generalizability. Generally, it provides partial support to the research hypotheses. The results should be interpreted with caution because of the cross-sectional survey, which is not sufficient for establishing a causal relationship. Another limitation of this study is the small and homogeneous sample; the low response rate of the physician sub-sample may have caused the loss of important information and differences. For more complex designs, including intervention studies and cohort designs, larger samples would be required and would allow for more sophisticated statistical analyses. It is important to draw attention to the fact that the sample does not include ethnic and cultural diversity.

Further research should consider experimental design, particularly for the development of interventions that may be more prone to participant effects, and in which cultural variation may impact patient-perceived acceptability and effectiveness.

## 8. Conclusions

There is now greater consideration than before that stressors may be important in predicting influence reactions in satisfaction outcomes. Our findings underline the view that dispositional mindfulness plays a role in helping employees improve their emotional job demands, satisfying emotional state, and mitigating reactions to organizational stress. The results appear particularly relevant in the specific case of nurses, whose contact with patients is an emotionally stressful context that can jeopardize well-being outcomes and affect the work performance. In line with the broader literature, findings suggest that interventions aimed at targeting mindfulness processes may also be more general, including quality of life. The results should be considered as a preliminary assessment for a study of broader interventions to evaluate which prevention programs can be implemented in favor of stressed nurses.

## Figures and Tables

**Figure 1 ijerph-17-03818-f001:**
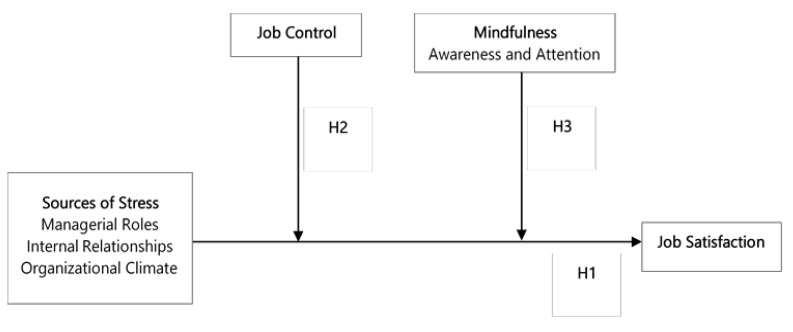
Tested conceptual model: job control and mindfulness as independent moderators of the relationship between sources of stress and job satisfaction. H, hypothesis.

**Figure 2 ijerph-17-03818-f002:**
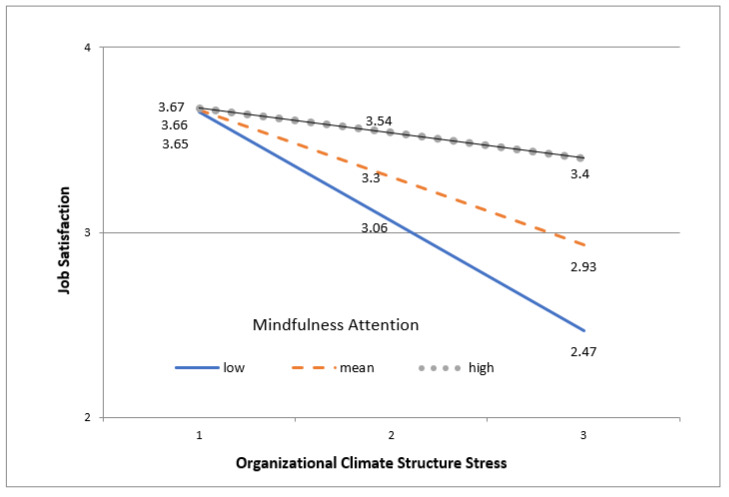
Interaction Between Climate (structure) and Mindfulness (attention) in the explanation of Variation in Job Satisfaction.

**Table 1 ijerph-17-03818-t001:** Sample characteristics (*n* = 148).

Variables	*n*	%
Schooling		
High school	70	47.3
Graduate	69	46.7
missing cases		9
Type of contract		
Long term	130	87.8
Fixed term	15	10.1
missing cases		3
Time		
Full time	144	97.3
Part-time	2	1.4
missing cases		2
Job shift		
Daily	52	35.1
Day/Night	54	36.5
No shift	37	25.0
missing cases		5

**Table 2 ijerph-17-03818-t002:** Study variables: descriptive statistics and bivariate correlations (*n* = 148).

Variables	M (SD)	1	2	3	4	5	6	7	8
1. Managerial role	4.24 (1.11)	1							
2. Climate Power	4.15 (1.12)	0.558 ***	1						
3. Climate Structure	4.15 (1.10)	0.532 ***	0.535 ***	1					
4. Internal Relationships	3.78 (1.08)	0.570 ***	0.530 ***	0.528 ***	1				
5. Mindful Awareness	6.49 (1.08)	0.058	−0.083	−0.128	−0.133	1			
6. Mindful Attention	4.05 (1.04)	−0.118	−0.280 **	−0.107	−0.193 *	0.529 ***	1		
7. Job Control	5.72 (1.08)	−0.147	−0.182 *	−0.084	−0.240 **	0.015	0.204 *	1	
8. Job Satisfaction	3.27 (1.14)	−0.207 *	−0.346 ***	−0.295 **	−0.160	0.030	0.228 **	0.285 **	1

* *p* < 0.05. ** *p* < 0.01. *** *p* < 0.001. M, Mean; SD, standard deviation.

**Table 3 ijerph-17-03818-t003:** Hierarchical regression analyses of socio-demographic variables, and factors affecting job satisfaction (using 1000 bootstraps) (*n* = 148).

Variables	Job Satisfaction					
	Model 1			Model 2		
	β	t	*p*	β	t	*p*
**Model 1:** Main effects of Socio-Demographic variables						
Age	−0.097	−0.934	0.352	−0.143	−1.548	0.124
Gender	−0.227	−2.628	<0.05	−0.266	−3.433	<0.01
Years	0.239	2.371	<0.05	0.215	2.449	<0.05
Working hours	−0.197	−2.349	<0.01	−0.123	−1.667	0.098
**Model 2:** Main effects of Psychological Construct						
Managerial Role				−0.059	−0.590	0.627
Climate Power				−0.166	−1.667	0.184
Climate Structure				−0.293	−2.819	<0.01
Relationship				0.131	1.384	0.279
Mindfulness Awareness				−0.093	−1.042	0.312
Mindfulness Attention				0.175	1.872	0.076
Job Control				0.224	2.817	<0.05
R^2^	0.133			0.393		
Adjusted R^2^	0.105			0.336		
Omnibus test of the regression	F (4, 124) = 4.768 **			F (7, 117) = 7.158 ***		

* *p* < 0.05. ** *p* < 0.01. *** *p* < 0.001.

**Table 4 ijerph-17-03818-t004:** Hierarchical regression analyses of the independent and interactive associations of sources of stress (FS—Climate, organizational), mindfulness Awareness, and job control (JC) with job satisfaction (using 1000 bootstraps) (*n* = 148).

Variables	Job Satisfaction								
	Model 1			Model 2			Model 3		
	β	t	*p*	β	t	*p*	β	t	*p*
	**Model 1**: Main effects								
Climate Structure	−0.275	−3.544	<0.01	−0.293	−3.670	<0.01	−0.293	−3.732	<0.01
Mind. Awareness	−0.009	−0.122	0.903	−0.015	−0.193	0.839	−0.015	−0.295	0.769
Job control	0.262	3.408	<0.01	0.252	3.242	<0.01	0.252	3.242	<0.01
				**Model 2**					
Climate structure × MIND				0.078	0.958	0.340			
							**Model 3**		
Climate structure × JC							0.060	0.771	0.442
R^2^	0.155			0.161			0.164		
Adjusted R^2^	0.138			0.137			0.135		
Omnibus test of the regression	*F* (3, 144) = 8.832 ***			*F* (1, 143) = 0.917			*F* (1, 142) = 0.595		

* *p* < 0.05. ** *p* < 0.01. *** *p* < 0.001.

**Table 5 ijerph-17-03818-t005:** Hierarchical regression analyses of the independent and interactive associations of sources of stress (FS—Climate, Organizational), mindfulness attention, and job control (JC) with job satisfaction (using 1000 bootstraps) (*n* = 148).

Variables				Job Satisfaction					
	Model 1			Model 2			Model 3		
	β	t	*p*	β	t	*p*	β	t	*p*
	**Model 1:** Main effects								
Climate Structure	−0.260	−3.407	<0.01	−0.352	−3.958	<0.01	−0.303	−3.934	<0.01
Mind. Attention	0.152	1.966	0.051	0.167	2.190	<0.05	0.166	2.164	<0.05
Job control	0.232	2.997	<0.05	0.212	2.775	<0.05	0.212	2.765	<0.05
				**Model 2**					
Climate structure × MIND				0.200	2.621	<0.05			
							**Model 3**		
Climate structure × JC							0.004	0.053	0.958
R^2^	0.177			0.215			0.215		
Adjusted R^2^	0.160			0.193			0.187		
Omnibus test of the regression	*F* (3,144) =10.352 ***			*F* (1, 143) = 6.870 **			*F* (1, 142) = 0.958		

* *p* < 0.05. ** *p* < 0.01. *** *p* < 0.001.

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
