# Peer review of "Mindfulness as a Protective Factor for Dissatisfaction in HCWs: The Moderating Role of Mindful Attention between Climate Stress and Job Satisfaction"

_ijerph, 2020, doi:10.3390/ijerph17113818_

Round 1

Reviewer 1 Report

HWCs or HRW do not explain their meaning the first time it is used in the text. Please correct it.

Why does nursing appear in the keywords when the sample is mixed? Please delete it.

The introduction needed a greater contextualization of the problem at the global, national and local levels, using quotes from international organizations that demonstrate the impact of the problem presented in the article. Please develop it in the text.

  1. Materials and Methods; Describe the type of study, the selection of the sample and the possible segregations that the sample presents. Please develop it in the text.

Total number of professionals at the center to calculate the samples. Please develop it in the text.

Bibliography justifies the presence of citations of more than 5 years existing articles of the same latest theme.

Appointments over 5 years 28/78

Author Contributions: complete this field, Please develop it in the text.

Reviewer 2 Report

When comparing with the previous version, this version of manuscript may have some improvement.  However, some key issues are still unclear, and they may need to be further clarified.  Here is my observation.

1.  Please don't use any abbreviation in the topic and also the abstract.

2.  As mentioned in previous comment, the authors should provide sufficient theories to support the significance of this study.

3.  For the part of mindfulness, I think the authors may need to illustrate how mindfulness practice can assist the employees to improve their well-being under adverse conditions.  Also, they should consider how they can accurately measure mindfulness practice in the analysis.  I remember that they focus on discussing dispositional mindfulness rather than mindfulness practice.  These two concepts are quite different.

4.  For the proposed model, I guess it should be used to test join moderating effects of job control and mindfulness.  If not, the authors should provide a clear explanation on how they examine the moderating role of job control and mindfulness.  When necessary, they need to revise all hypotheses.

5.  Please report the alphas of each measure in appropriate areas.

6.  For the analysis of moderating effects, I find the the equations were different from other related analyses.  It seems necessary to revise the analysis of moderating effects.

7.  Due to Point 6, I am unable to comment the part of discussion and implication.

    Hope that my observation is useful, and thanks for giving me this opportunity to review this manuscript

Author Response

This manuscript is a resubmission of an earlier submission. The following is a list of the peer review reports and author responses from that submission.

Round 1

Reviewer 1 Report

The manuscript is too long in the introduction, try a shorter approach. In the Bibliography there are very old citations, please update with more recent citations.

In line with these theories on job control [31, 32, 26, 38] researchers showed that
96 apparently, there
97 is a consistent evidence that high levels of work control are associated with low levels of stress-related
98 outcomes [36,39].

Some quotes are wrong in the text like this example.
156 3. Study aim and hypotheses

Reduce the section to only the hypotheses, too much justification would eliminate from line 157-169 They add nothing and create a bit of confusion for the reader.

4. Materials and Methods

From the method I would like to clarify if the data are original or as they appear to be from a larger previous study. If so, they contribute again that these authors did not previously publish.
All participants were asked to complete a questionnaire on a voluntary basis (100% response
212 rate). I am not aware of any study with 100% response rate and in its own table 1 it indicates lost.

5.3. 317 Hypothesis tests

I would like them to be clearer in the explanation of hypothesis 2, it is not clear to me why they reject it-

They should update appointments 59/117 are over 5 years old. Except for very specific quotes from referents in the matter, you should not abuse obsolete quotes in the articles.

Reviewer 2 Report

This topic is not a novel topic in Western societies, and I just feel that this manuscript may add some additional information regarding the moderating role of dispositional mindfulness in relation to workplace stress.  I think the authors may need to contribute more input on improving the content of this manuscript.  

Particularly, the authors require to add more theoretical literature regarding the significance of this topic.  In the meantime, they need to further elaborate how this disposition affects nurses' cognition and behaviours when confronting the adverse conditions.  Furthermore, what is the practical implication of results for stress intervention among the nurses, managerial staff and the organisation respectively.

Maybe, the authors can add more innovative ideas in order to enhance the chances of acceptance for publications in referred journals.

Many thanks for giving me to review this manuscript.  Good luck!